# Position: Ignoring Hyperparameter Tuning Costs Misleads the Development of Efficient RL Algorithms

## Abstract

The performance of reinforcement learning (RL) algorithms is often benchmarked without accounting for the cost of hyperparameter tuning, despite its significant practical impact. In this position paper, we argue that such practices distort the perceived efficiency of RL methods and impede meaningful algorithmic progress. We formalize this concern by proving a lower bound showing that tuning $m$ hyperparameters in RL necessarily induces an exponential $\exp(m)$ blow-up in the sample complexity or regret, in stark contrast to the linear $O(m)$ overhead observed in supervised learning. This highlights a fundamental inefficiency unique to RL. To address this, we propose evaluation protocols that account for the number and cost of tuned hyperparameters, enabling fairer comparisons across algorithms. Surprisingly, we find that once tuning cost is included, elementary algorithms can outperform their successors with more sophisticated design. These findings call for a shift in how RL algorithms are benchmarked and compared, especially in settings where efficiency and scalability are critical.

## 1 Introduction

While lacking a universally agreed definition, *hyperparameters* are broadly considered parameters that are set prior to running an algorithm and remain fixed throughout its execution. Examples include the step size in optimization, regularization coefficients, neural network architecture choices (e.g., depth, width), and activation functions. Although theoretical guidelines exist for some of these parameters (e.g., $\Theta(1/\sqrt{T})$ step size in stochastic gradient descent), practical deployments typically require manual or automated **hyperparameter optimization (HPO)** to identify the problem-specific optimal values. Due to the non-differentiable and sometimes discrete nature of this search space, HPO is usually done via grid search or derivative-free optimization over a combinatorial search space.

When it comes to reinforcement learning (RL), algorithm performance is notoriously sensitive to hyperparameter choices [Patterson et al., 2023, Eimer et al., 2023, Adkins et al., 2024, Obando-Ceron et al., 2024]. Alarmingly, some studies have even reported practices of tuning random seeds as hyperparameters to overfit to public benchmarks[Henderson et al., 2018]. Over the past decades, the number of hyperparameters in RL algorithms has steadily increased. For example, the original DQN algorithm [Mnih et al., 2015] required selecting 16 hyperparameters, while Rainbow [Hessel et al., 2018] introduced 25. This rising trend is illustrated in Figure 1. Numerous works have acknowledged this phenomenon and provided practical guidelines for selecting and evaluating hyperparameters in RL [Franke et al., 2021, Eimer et al., 2022, 2023, Patterson et al., 2024].

In this paper, we take on a more quantitative perspective and pose the following central question:

Submitted to 39th Conference on Neural Information Processing Systems (NeurIPS 2025). Do not distribute.

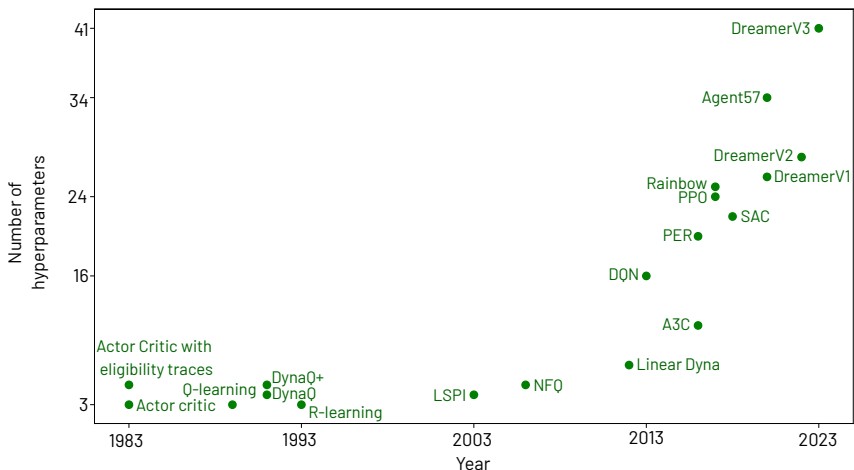

Figure 1: The number of hyperparameters in reinforcement learning algorithms proposed over the last decades [Adkins et al., 2024].

**What is the statistical cost of HPO in online RL, and how should it be quantified?**

We define this statistical cost as the additional sample complexity or regret incurred during HPO. In Section 3, we show that tuning $m$ hyperparameters results in a sample complexity overhead of $\Theta(\exp(m))$ in RL—a stark contrast to the $O(m)$ overhead typically observed in supervised learning (SL). This exponential cost is especially problematic in settings where real-world interaction is expensive, simulators are unavailable, or large-scale offline datasets are lacking.

Building on this analysis, we propose evaluation protocols that explicitly account for HPO overhead. These enable fairer comparisons across algorithms and support practical questions such as:

- Is Algorithm A truly more data-efficient than Algorithm B?
- Which hyperparameters are worth tuning in a given deployment?

We explore these use cases in Section 4. Ultimately, we advocate for a new goal in RL research: the development of **parameter-free RL algorithms** that minimize or eliminate the need for hyperparameter tuning.

## 2 Related work

Early works such as Henderson et al. [2018], Machado et al. [2018] emphasized reproducibility challenges in RL, attributing much of the variance to opaque or inconsistent hyperparameter settings. Subsequent studies examined hyperparameter sensitivity more systematically across benchmark environments, introducing new sensitivity metrics and evaluation methods [Eimer et al., 2022, 2023, Adkins et al., 2024]. Others advocated for AutoML solutions to automate HPO in RL [Franke et al., 2021, Eimer et al., 2023], or proposed benchmarks requiring shared hyperparameter configurations across tasks [Patterson et al., 2024]. However, these works primarily focus on empirical sensitivity or tuning practices. None explicitly quantify the *statistical cost* of HPO in RL or examine its impact on algorithm comparison. Our paper fills this gap, offering a theoretical framework that complements these empirical efforts.

On the theoretical side, the cost of tuning has been explored under the lens of *online model selection*, where the goal is to choose the best base algorithm from a finite set [Agarwal et al., 2017, Abbasi-Yadkori et al., 2020, Ghosh et al., 2020, Chatterji et al., 2020, Bibaut et al., 2020, Foster et al., 2020, Lee et al., 2020, Wei et al., 2022]. Since tuning $m$ hyperparameters often corresponds to evaluating $M = O(\exp(m))$ configurations, HPO effectively reduces to a model selection problem over an exponentially large set. Approaches include FTRL-based methods (e.g., EXP4 [Odalric and Munos, 2011], Corral [Agarwal et al., 2017], Tsallis-INF [Arora et al., 2020]) and regret-balancing schemes [Pacchiano et al., 2020, Cutkosky et al., 2021]. A common limitation is the requirement for known regret bounds for each base algorithm—a barrier in real-world RL where such guarantees are unavailable.

A recent exception is Dann et al. [2024], who propose an algorithm that competes against the *realized regret* of the best base model without requiring candidate regret bounds. They achieve regret $\widetilde{O}(d_T^\star M\sqrt{T} + (d_T^\star)^2\sqrt{MT})$, where $d_T^\star\sqrt{T}$ is the presumed regret of the best base model. On the other hand, the best known lower bound takes the form of $\Omega((d_T^\star)^2\sqrt{T})$ [Marinov and Zimmert, 2021], implying that recovering the regret rate of the best base learner is in general not possible. Interestingly, prior work in this community has emphasized dependence on $d^\star$ over $M$—in contrast to our focus on the exponential dependence in $M$ arising from HPO.

Designing **parameter-free** algorithms directly is an alternative. This goal has been extensively pursued in optimization [Defazio and Mishchenko, 2023, Carmon and Hinder, 2022, Ivgi et al., 2023, Cutkosky et al., 2024, Khaled and Jin, 2024] and online learning [Orabona and Pál, 2018, Cutkosky and Orabona, 2018, van der Hoeven et al., 2020]. In RL, however, work on parameter-free algorithms is scarce. Recent theoretical contributions include algorithms that adapt to unknown reward scales or state-space sizes [Chen and Zhang, 2023, 2024, Chen et al., 2024]. Empirical studies like Yu et al. [2021] also explore adaptive hyperparameter schemes, though without the explicit goal of parameter-free design.

# 3 The Statistical Cost of Hyperparameter Optimization (HPO)

Historically, HPO has received minimal attention in both algorithm design and theoretical analysis because its cost is modest in classic settings like SL. In those domains, data splitting strategies such as cross-validation yield efficient HPO procedures. We begin by formalizing this baseline in SL and contrast it with the RL setting.

## 3.1 HPO Cost in Supervised Learning

Let $D = \{(x, y)_i\}_{i=1}^N$ be an i.i.d. dataset sampled from a distribution $P$. Assume a supervised learning algorithm $\mathcal{A}_\theta$ is parameterized by hyperparameter $\theta$ and returns a predictor in a hypothesis class $\mathcal{F}$.

**Definition 3.1** (PAC-learner in SL). *A learner $\mathcal{A}$ is a PAC-learner if, for all $\delta \in (0, 1)$, with probability at least $1 - \delta$ over draw of dataset D, we have:*

$$\mathbb{E}_{(x,y)\sim P}[(\mathcal{A}(D)(x) - y)^2] - \min_{f\in\mathcal{F}}\mathbb{E}[(f(x) - y)^2] \leq \epsilon(N, \delta).$$

*where $\epsilon(N, \delta)$ denotes the optimality gap.*

Many SL algorithms achieve an optimality gap of the form:

$$\epsilon(N, \delta) = O\left(\sqrt{\frac{C_\mathcal{F}\log(1/\delta)}{N}}\right), \tag{1}$$

where $C_\mathcal{F}$ denotes the complexity of $\mathcal{F}$. An alternative metric to quantify learning efficiency is the **sample complexity**, i.e. the number of samples it requires to reach a certain optimality gap. For instance, (1) would translate to a sample complexity of

$$N(\epsilon, \delta) = O\left(\frac{C_\mathcal{F}\log(1/\delta)}{\epsilon^2}\right). \tag{2}$$

In supervised learning, HPO can be implemented using a simple data-splitting approach (Algorithm 1), where models are trained on the training data and hyperparameters are selected based on losses on the validation data:

**Theorem 3.2** (PAC Guarantee of SL HPO). *With probability $1 - \delta$, Algorithm 1 returns $\hat{f}$ satisfying:*

$$\mathbb{E}[(\hat{f}(x) - y)^2] - \min_{f\in\mathcal{F}}\mathbb{E}[(f(x) - y)^2] \leq 2\min_{\theta\in\Theta}\epsilon_\theta(N/2, \delta/|\Theta|) \approx \tilde{O}\left(\min_{\theta\in\Theta}\sqrt{\frac{\log|\Theta|C_{\mathcal{F}_\theta}}{N}}\right).$$

In other words, Algorithm 1 is itself a PAC-learner with no hyperparameter and a sample complexity $\log|\Theta|$ times that of the best base learner. This $\log|\Theta|$ comes from a union bound over $\Theta$ and is

---
**Algorithm 1** HPO via Data Splitting in SL

---
1: **Input:** Dataset $D$ of size $N$, hyperparameter set $\Theta$, learning algorithm $\mathcal{A}_\theta$
2: Split $D$ into $D_{train}$ and $D_{val}$ of size $N/2$
3: **for** $\theta \in \Theta$ **do**
4:     Train $f_\theta = \mathcal{A}_\theta(D_{train})$
5:     Evaluate $\epsilon_\theta = \frac{2}{N} \sum_{(x,y) \in D_{val}} (f_\theta(x) - y)^2$
6: **end for**
7: Return $\hat{\theta} = \arg\min_{\theta \in \Theta} \epsilon_\theta$, model $f_{\hat{\theta}}$

---

negligible in the big data regime ($N \gg \log|\Theta|$). This benign scaling is possible mainly due to data sharing across different hyperparameters, i.e. $\mathcal{A}_{\theta_1}$ and $\mathcal{A}_{\theta_2}$ can use the same training and validation data. As a result, Algorithm 1 is *modular* — it only requires **black-box access** to the learner, making it applicable to any learner $\mathcal{A}$. As a result of how effortless HPO is in SL, modern deep learning has evolved under nearly no selection pressure towards having fewer hyperparameters. However, it turns out to be catastrophic for HPO in reinforcement learning.

## 3.2 HPO Cost in Online Reinforcement Learning

In RL, we no longer have fixed datasets nor trivial validation procedures. Consider an online PAC-RL setting, where the agent interacts with an unknown environment with the goal of finding a near-optimal policy in as few episodes as possible.

**Definition 3.3** (PAC-agent in Online RL). *Given an MDP $\mathcal{M}$, a PAC-agent is defined as a learning agent that, with probability at least $1 - \delta$, after interacting with $\mathcal{M}$ for $T$ episodes, returns a policy $\hat{\pi}$ that satisfies*

$$\max_\pi V_{\mathcal{M}}^\pi - V_{\mathcal{M}}^{\hat{\pi}} \le \epsilon(T, \delta),$$

*for some function $\epsilon : \mathbb{N} \times (0, 1) \to \mathbb{R}^+$.*

Naively adapting Algorithm 1 to RL involves training a policy for each $\theta \in \Theta$ using $T/|\Theta|$ episodes (since data can no longer be shared seamlessly across different agents and will instead be split evenly), then evaluating each and selecting the best. This yields the following result:

**Theorem 3.4** (PAC Guarantee in RL). *With probability $1 - \delta$, this procedure returns $\hat{\pi}$ satisfying:*

$$\max_\pi V^\pi - V^{\hat{\pi}} \le \min_{\theta \in \Theta} \epsilon_\theta(N/|\Theta|, \delta/|\Theta|) \approx \tilde{O}\left(\min_{\theta \in \Theta} \sqrt{\frac{|\Theta| C_{\mathcal{F}_\theta}}{T}}\right).$$

The exponential size of $|\Theta| := M = O(\exp(m))$ implies that tuning $m$ hyperparameters incurs an $\Omega(\exp(m))$ overhead. To make things worse, we can in fact show that the above is not just a weakness of this naive algorithm but rather a necessary cost for any black-box hyperparameter tuning algorithm:

**Theorem 3.5** (Black-box HPO is inefficient in RL). *Given any base agent $\mathcal{A}$ with a hyperparameter set $\Theta$ and denote $\epsilon_\theta(T, \delta)$ the optimality gap function corresponding to hyperparameter $\theta$. No black-box HPO algorithm can return a policy $\hat{\pi}$ that satisfies*

$$\max_\pi V^\pi - V^{\hat{\pi}} \le \min_{\theta \in \Theta} \epsilon_\theta(2T/|\Theta|, 1/2)$$

*with probability greater than $1/2$.*

This lower bound builds on two key observations: First, no data sharing between base agents is possible in online RL without algorithm specific structures, which is unobtainable in the black-box setting. Thus, the total budget of $T$ episodes must be split between $|\Theta|$ different base agents. Second, no base agents can be dropped prematurely without additional assumption on $\epsilon_\theta(T, \delta)$ beyond monotonicity, e.g. a base agent that starts off perform poorly could potentially catch up and becomes the best after some time. Therefore, there is in fact no strategy that guarantees to be better than spending $T/|\Theta|$ episodes on each base agent in the worst case.

Subsequently, the regret to sample complexity reduction implies a similar lower bound on the regret:

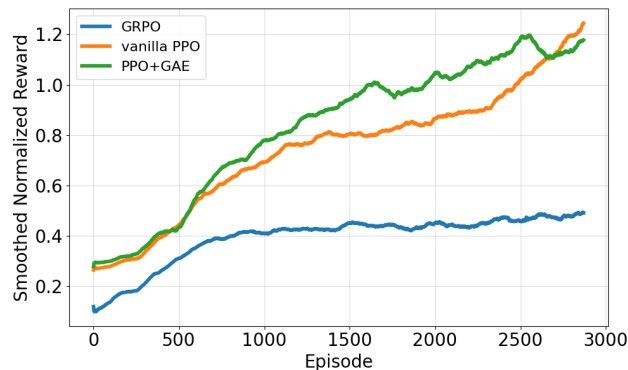

Figure 2: A typical comparison between different RL algorithms based on the performance with the best hyperparameter configuration for each algorithm. A plausible conclusion one may draw from a plot like this is `PPO+GAE` > vanilla `PPO` > `GRPO`.

**Corollary 3.6** (Regret Lower Bound). *Given any base agent $\mathcal{A}$ with a hyperparameter set $\Theta$ and denote $Reg_\theta(T, \delta)$ the regret corresponding to hyperparameter $\theta$, where regret is defined as the cumulative optimality gap during the execution of the algorithm, $\sum_{t=1}^{T}(\max_\pi V^\pi - V^{\hat{\pi}_t})$. Then, no black-box HPO algorithm can achieve a regret bound better than*

$$\min_{\theta \in \Theta} \frac{|\Theta|}{2} Reg_\theta(2T/|\Theta|, 1/2)$$

*with probability more than $1/2$.*

Notice that the lower bound in Theorem 3.5 matches with the upper bound in Theorem 3.4, implying that the naive strategy of splitting data equally across all hyperparameters is in fact optimal for the pure exploration problem. When it comes to regret, Corollary 3.6 complements the existing lower bound of Marinov and Zimmert [2021]. Yet, there is still a gap between the best known upper bound of Dann et al. [2024] and both lower bounds left for future research to resolve. Nevertheless, our lower bounds are sufficient to show that, unlike SL, the cost of hyperparameter tuning in RL is *multiplicative* rather than logarithmic.

However, most RL papers ignore this cost in reporting algorithm performance, often presenting results for the best-tuned hyperparameter configuration while omitting the number of trials or total samples used. An example of such practices is given in Figure 2. Such practices give an unfair advantage to complex algorithms with more tunable hyperparameters. A model with twice the learning speed but ten times the tuning burden may still be presented as superior. This dynamic skews empirical comparisons and hampers progress toward scalable, real-world RL. In the following sections, we show how incorporating HPO cost into evaluation can reshape algorithm rankings, and how elementary algorithms can outperform sophisticated baselines when tuning is properly accounted for.

## 4 Proposed Metrics

Despite the lack of an optimal hyperparameter optimization (HPO) algorithm for RL (in terms of regret), it remains essential to measure and compare the learning efficiency of RL algorithms in a way that fairly incorporates the cost of tuning. In this section, we adopt an *optimistic* yet principled approach: we use the theoretical lower bounds established in Section 3 as a guideline to construct practical evaluation metrics. These metrics serve to assess the performance of RL algorithms while explicitly penalizing for the number of hyperparameters being tuned.

We introduce two core metrics: *Effective Sample Complexity* and *Effective Area Under the Curve*. These metrics aim to mirror real-world deployment scenarios where tuning is costly, and help distinguish algorithms that are truly efficient from those that merely perform well under exhaustive hyperparameter search.

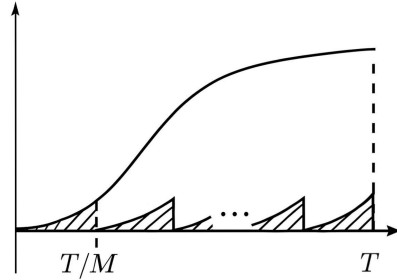

Figure 3: An illustration of the Effective AUC, represented by the shaded area under the curve.

**Definition 4.1** (Effective Sample Complexity (SC)). *Let $\mathcal{A}$ be a reinforcement learning algorithm with a hyperparameter set $\Theta$ and let $\theta \in \Theta$ be given. The **Effective Sample Complexity** required to reach an optimality threshold $\epsilon$ is defined as:*

$$|\Theta| \times T_\theta(\epsilon), \tag{3}$$

*where $T_\theta(\epsilon)$ is the number of episodes needed for $\mathcal{A}_\theta$ to produce a policy $\hat{\pi}$ such that $\max_\pi V^\pi - V^{\hat{\pi}} \leq \epsilon$.*

This metric captures the cost of HPO in the number of environment interactions to find a near-optimal policy. Notice that (3) resembles the matching upper and lower bounds of Theorem 3.5 and Theorem 3.4 and the multiplicative $|\Theta|$ factor reflects the fundamental inefficiency identified in our theoretical analysis. Importantly, we assume the best configuration is selected in hindsight, making this a lower bound on true tuning cost. In our experiments, the threshold $\epsilon$ corresponds to the 90th percentile of episodic returns aggregated over all configurations and algorithms.

While sample complexity measures the data needed to achieve competent performance, many real-world applications also care about cumulative reward during the learning process. Thus, we introduce a regret-based metric:

**Definition 4.2** (Effective Area Under the Curve (AUC)). *Let $\mathcal{A}$ be an RL algorithm with hyperparameter set $\Theta$ and let $\theta \in \Theta$ be given. The **Effective AUC** over $T$ episodes is defined as:*

$$|\Theta| \times \sum_{t=1}^{T/|\Theta|} V^{\pi_{\theta,t}}, \tag{4}$$

*where $V^{\pi_{\theta,t}}$ is the expected reward of the policy at episode $t$ when using configuration $\theta$.*

Intuitively, (4) measures the cumulative rewards achieved by each configuration over a period of $T/|\Theta|$ episodes, then multiplying it by $|\Theta|$, as illustrated in Figure 3. This is derived directly from our lower bound in Corollary 3.6, and should be viewed as *optimistic*, in the sense that this is the least amount of regret one would suffer by calling an online model selection algorithm for hyperparameter tuning. Notice that the naive data splitting framework in Algorithm 1 would incur a $O(T)$ regret in the worst case, because there is no guarantee on how much more regret a suboptimal hyperparameter would incur comparing to the optimal one.

Taken together, these metrics allow us to rethink what it means for an RL algorithm to be efficient. Rather than asking "how well does this algorithm perform after tuning?", we ask "how much reward and performance is achievable if we must account for the tuning effort?"

## 5 Experiment

We evaluate the proposed metrics on a suite of continuous control tasks from the MuJoCo benchmark, including Hopper, Ant, Swimmer, HalfCheetah, and Walker2d. Our goal is twofold: (1) guide practitioners in making tuning decisions under practical constraints, and (2) assess how tuning overhead alters algorithm comparisons.

| Algorithm | Actor LR | Critic LR | Entropy Coef ($\tau$) | GAE $\lambda$ | $M$ |
|---|---|---|---|---|---|
| GRPO | {1e-5, 3e-5, 1e-4} | NA | {1e-3, 1e-2, 1e-1} | NA | $3^2$ |
| Vanilla PPO | {1e-5, 3e-5, 1e-4} | {1e-4, 3e-4, 1e-3} | {1e-3, 1e-2, 1e-1} | NA | $3^3$ |
| PPO+GAE | {1e-5, 3e-5, 1e-4} | {1e-4, 3e-4, 1e-3} | {1e-3, 1e-2, 1e-1} | {0.5, 0.7, 0.9} | $3^4$ |
| PPO+ADVN | {1e-5, 3e-5, 1e-4} | {1e-5, 1e-4, 1e-3} | {1e-3, 3e-3, 1e-2} | {0.3, 0.5, 0.7} | $3^4$ |

Table 1: Hyperparameter set for each algorithm.

| Number of HPs tuned | 4 | 3 | 2 | 1 |
|---|---|---|---|---|
| PPO+GAE (AUC) | actorlr, criticlr, $\tau$, $\lambda$ | criticlr, $\tau$, $\lambda$ | $\tau$, $\lambda$ | $\lambda$ |
| PPO+ADVN (AUC) | actorlr, criticlr, $\tau$, $\lambda$ | criticlr, $\tau$, $\lambda$ | criticlr, $\lambda$ | $\lambda$ |
| PPO+GAE (SC) | actorlr, criticlr, $\tau$, $\lambda$ | actorlr, $\tau$, $\lambda$ | $\tau$, $\lambda$ | $\lambda$ |
| PPO+ADVN (SC) | actorlr, criticlr, $\tau$, $\lambda$ | actorlr, $\tau$, $\lambda$ | actorlr, $\lambda$ | $\lambda$ |

Table 2: Hyperparameters tuned for PPO+GAE and PPO+ADVN.

**Algorithms**  We focus on four policy gradient algorithms: Group Relative Policy Optimization (GRPO), vanilla Proximal Policy Optimization (PPO), PPO+Generalised Advantage Estimator (PPO+GAE), and a variant of PPO that performs Advantage per-minibatch zero-mean normalization (PPO+ADVN).

All algorithms optimize the same clipped surrogate PPO objective:

$$L(\theta) = \hat{\mathbb{E}}_t \left[ \min \left( r_t(\theta) \hat{A}_t, \, \text{clip}(r_t(\theta), 1 - \epsilon, 1 + \epsilon) \hat{A}_t \right) \right], \qquad (5)$$

Each algorithm differs in how $\hat{A}_t$ is computed.

- GRPO: $\hat{A}_t^{GRPO} = \frac{R_t - \text{mean}(R_t)}{\text{std}(R_t)}$

- Vanilla PPO: $\hat{A}_t^{PPO} = R_t - V(s_t)$

- PPO+GAE: $\hat{A}_t^{GAE} = \delta_t + (\gamma\lambda)\delta_{t+1} + \cdots + (\gamma\lambda)^{T-t+1}\delta_{T-1}$, where $\delta_t = r_t + \gamma V(s_{t+1}) - V(s_t)$.

- PPO+ADVN: $\hat{A}_t^{ADVN} = \frac{\hat{A}_t^{GAE} - \text{mean}(\hat{A}_t^{GAE})}{\text{std}(\hat{A}_t^{GAE})}$

In particular, GRPO does not use a value network and therefore only has two hyperparameters: actor learning rate (LR) and entropy regularizer coefficient ($\tau$). Vanilla PPO uses a value network to help with the estimation of the advantage function and thus have the critic learning rate (LR) as an additional hyperparameter. Both PPO+GAE and PPO+ADVN additionally have the GAE hyperparameter ($\lambda$). The corresponding hyperparameters and their values are listed in Table 1. For each hyperparameter configuration, we run 10 independent trials per environment, with each trial consisting of 3,000,000 timesteps.

**Preprocessing**  Since raw reward scales differ across environments, we normalize each trajectory's return $R(\tau)$ to the range $[0, 1]$ using per-environment quantile normalization:

$$\bar{R}(\tau) = \frac{R(\tau) - p_5(e)}{p_{95}(e) - p_5(e)}, \qquad (6)$$

where $p_5(e)$ and $p_{95}(e)$ are the 5% and 95% quantiles of returns in environment $e$. This ensures fair metric comparison across tasks. We then average the reward-vs-T curve across 10 trials to get a single curve per (environment, algorithm, hyperparameter) tuple and calculate the effective SC and effective AUC using these curves.

## 5.1 Use Case I: Choosing Hyperparameters to Tune

Consider a practitioner deploying RL in a new task similar to MuJoCo. They face a practical question: which hyperparameters should be tuned, and which can be fixed based on prior knowledge

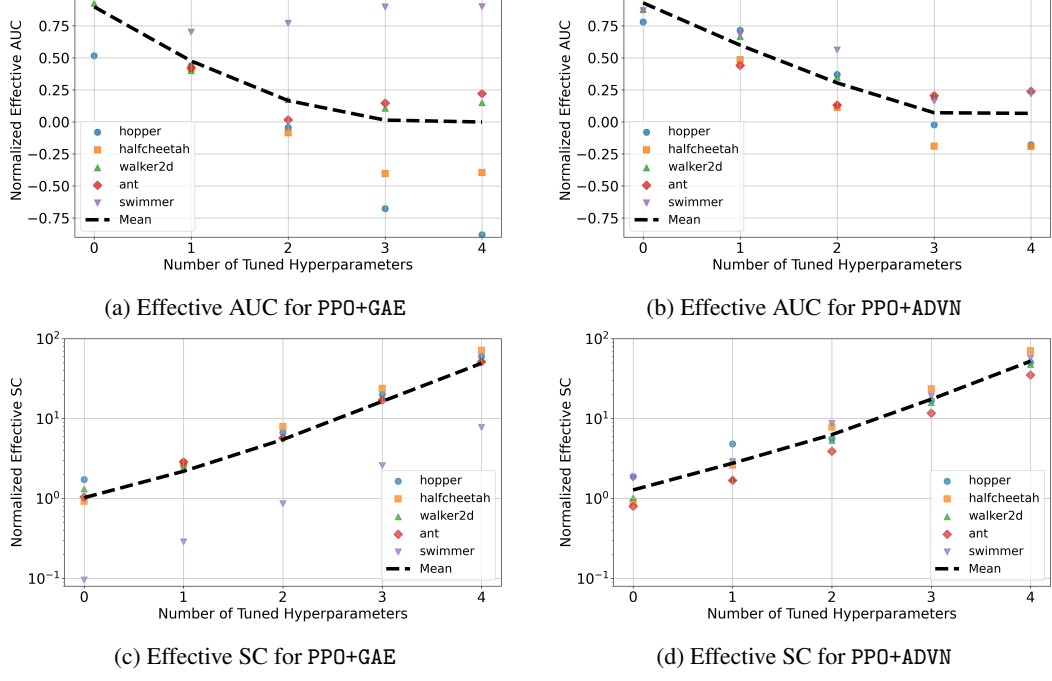

Figure 4: Normalized effective AUC&SC vs. number of tuned hyperparameters.

or auxiliary environments? Tuning all hyperparameters may yield the best result in hindsight, but it also incurs exponential cost. We simulate this setting using `PPO+GAE` and `PPO+ADVN`, both with 4 hyperparameters.

For each value $m = 0, 1, 2, 3, 4$, we search over all combinations of hyperparameters where $m$ of them can vary across environments while the other $(4 - m)$ are fixed across all environments. This procedure results in a sequence of hyperparameter spaces of increasing size: $M = 1, 3^1, 3^2, 3^3$ and $3^4$, corresponding to zero, one, two, three, and four tunable hyperparameters.

For each combination, we measure the effective AUC (respectively SC) in each environment. To compare across different environments, we normalize all effective AUC (respectively SC) values using their quantiles among the full-grid configurations in each environment, similar to how rewards are normalized, ensuring that easy environment and hard environment are weighted equally. Considering that the effective SC can vary significantly across configurations within the same environment, we apply a different normalization scheme for it:

$$\text{Normalized effective AUC} = \frac{AUC - p_{5,\text{AUC}}(e)}{p_{95,\text{AUC}}(e) - p_{5,\text{AUC}}(e)} \tag{7}$$

$$\text{Normalized effective SC} = \frac{SC}{p_{5,\text{SC}}(e)} \tag{8}$$

We then calculate the average normalized effective AUC (respectively SC) across all five environments. The hyperparameter configuration that achieves the highest average normalized effective AUC (respectively SC) is chosen as the optimal setup for its specific $m$. In other words, the optimal configuration for each $m$ tells us which hyperparameter should be fixed across environments and at what value, while the other hyperparameters should be tuned per environment. The hyperparameters that are tuned for each $m$ is shown in Table 2. The performances of the best configuration for each $m$ is shown in figure 4, across both algorithms and metrics. The dashed line represents the mean normalized effective AUC across environments at each $M$, highlighting overall trends.

Figure 4 shows a consistent trend: allowing more tunable hyperparameters does not always improve performance. In fact, both effective AUC and SC typically degrade with additional tuning flexibility. This suggests that in practice, a judicious selection of one or two hyperparameters to tune can outperform more complex tuning setups, particularly when the tuning budget is constrained.

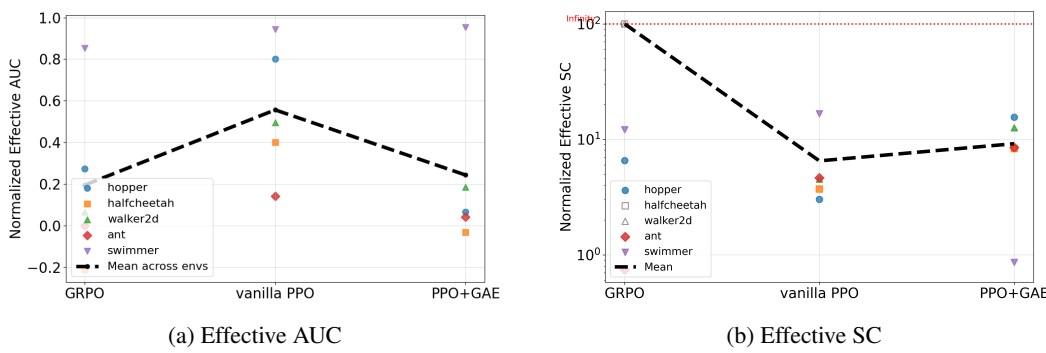

|                          |                          |
| :----------------------: | :----------------------: |
| (a) Effective AUC        | (b) Effective SC         |

Figure 5: Normalized Effective AUC/SC vs different algorithms.

## 5.2 Use Case II: Fair Comparison between RL Algorithms

Next, we use our metrics to reevaluate algorithm comparisons. We focus on three algorithms of increasing complexity: `GRPO` (2 hyperparameters), vanilla `PPO` (3 hyperparameters), and `PPO+GAE` (4 hyperparameters).

In standard RL benchmarking, algorithms are often compared based on the performance of their best hyperparameter configuration. As shown in Figure 2, this naive evaluation favors `PPO+GAE` in both AUC and sample complexity.

However, once we adjust for tuning cost via normalized effective AUC and SC (Figure 5), the ranking changes dramatically. Vanilla PPO consistently outperforms both `GRPO` and `PPO+GAE`, achieving a better balance between performance and tuning overhead. Interestingly, `GRPO`, despite having the fewest hyperparameters, performs the worst under both metrics and even fails to meet the 90% performance threshold in two environments, leading to infinite sample complexity.

## 6 Conclusion and Discussion

This work draws attention to a pervasive blind spot in reinforcement learning research: the statistical and computational cost of hyperparameter optimization. While supervised learning tolerates a modest $\log(|\Theta|)$ tuning overhead, RL suffers an exponential penalty, both in theory and in practice. Our results show that ignoring this cost leads to misleading conclusions in algorithm comparisons and suboptimal design decisions.

To address this, we introduce two metrics—*Effective Sample Complexity* and *Effective AUC*—that quantify an algorithm's learning efficiency while accounting for the cost of hyperparameter tuning. These metrics, grounded in theoretical lower bounds, reveal that many popular RL algorithms are less efficient than simpler alternatives once tuning cost is considered.

Empirical findings on MuJoCo tasks demonstrate that:

• Many hyperparameters are not worth tuning; their cost outweighs their benefit.

• Algorithms with more complex design do not always outperform simpler baselines when fair comparisons are made.

We advocate for a shift in evaluation practices. RL research should routinely include HPO-aware metrics and prioritize the development of **parameter-free algorithms** that minimize or eliminate tuning altogether. Only then can we build RL systems that are scalable, robust, and ready for real-world deployment.

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
