# OpenReview forum: "Position: Ignoring Hyperparameter Tuning Costs Misleads the Development of Efficient RL Algorithms"
_NeurIPS.cc/2025/Position_Paper_Track — Submitted to NeurIPS 2025 Position Paper Track_

### Official Review · Reviewer_kUYR · 2025-07-12

**Significance:** 3
**Presentation:** 2
**Rating:** 4
**Confidence:** 2

**Summary:**

This position paper argues that benchmarking RL algorithms should take into account the cost of hyperparameter tuning. Specifically, the authors demonstrate that the statistical cost of tuning $m$ hyperparameters in RL is significantly higher (e.g., exponential in RL compared to linear in supervised learning). They further propose two metrics to quantify an algorithm’s learning efficiency while incorporating hyperparameter tuning costs, and observe that many popular RL algorithms are less efficient than simpler alternatives. Based on these findings, the authors advocate for the development of parameter-free RL algorithms.

**Strengths:**

* The position of this paper is interesting and may hold value for the RL community.
* The argument is well-supported by both theoretical and empirical analyses.

**Weaknesses:**

* The authors mention the high cost of hyperparameter tuning in online RL settings (lines 34–39). However, it is unclear whether their position also applies to offline RL.
* Is developing parameter-free RL algorithms a feasible goal?
* Could the authors provide a more detailed derivation of the proof in Section 3?
* The writing and structure of the paper could be improved.

**Questions:**

Please see weaknesses.

**Alternative Position:**

No

**Author Identification:**

No.

**Context:**

2

**Discussion:**

3

**Ethics:**

["NO or VERY MINOR ethics concerns only"]

**Position:**

Yes, the paper argues for or against a position related to machine learning.

**Support:**

3

**Thoroughness:**

2

---

### Official Review · Reviewer_4UpD · 2025-07-30

**Significance:** 3
**Presentation:** 3
**Rating:** 6
**Confidence:** 3

**Summary:**

This position paper argues that current benchmarking practices in reinforcement learning (RL) overlook the statistical cost of hyperparameter tuning, which leads to misleading claims about algorithm efficiency. The authors formalize this concern by proving that tuning \( m \) hyperparameters in online RL incurs a sample complexity overhead of \( \Theta(\exp(m)) \), in stark contrast to the \( O(\log M) \) or \( O(m) \) overhead in supervised learning. They propose two tuning-aware evaluation metrics — **Effective Sample Complexity** and **Effective AUC** — to penalize algorithms that rely heavily on hyperparameter search. Empirical results demonstrate that when these metrics are applied, simpler algorithms with fewer tunable parameters can outperform more sophisticated alternatives. The paper concludes by advocating for a shift toward parameter-free or tuning-efficient RL algorithm design.

**Strengths:**

The paper clearly argues that ignoring hyperparameter tuning costs in reinforcement learning (RL) can lead to misleading comparisons between algorithms. It supports this argument with both theory—showing that tuning even a few hyperparameters can greatly increase sample complexity—and experiments on MuJoCo tasks. The proposed metrics (Effective Sample Complexity and Effective AUC) are simple and useful for comparing algorithms more fairly. The experiments are well designed and show that simpler methods can sometimes outperform complex ones when tuning costs are considered. This is a relevant and important topic for the NeurIPS community, where benchmarking practices often focus too much on best-case results.

**Weaknesses:**

The paper does not consider or discuss alternative viewpoints. For example, some may argue that reporting best-tuned results is still useful for comparing potential performance, especially if all methods are tuned equally. The authors also do not explore cases where tuning costs can be shared across tasks (e.g., via meta-learning or transfer HPO), which might reduce the impact of their main claim.

**Questions:**

How do the authors view the role of transfer or meta-learning approaches that reuse tuning knowledge across tasks—could these offer a practical way to reduce the effective tuning cost, and if so, how would that fit into their evaluation framework?

**Alternative Position:**

No

**Author Identification:**

No.

**Context:**

3

**Discussion:**

3

**Ethics:**

["NO or VERY MINOR ethics concerns only"]

**Position:**

Yes, the paper argues for or against a position related to machine learning.

**Support:**

3

**Thoroughness:**

3

---

### Official Review · Reviewer_oswF · 2025-08-07

**Significance:** 4
**Presentation:** 3
**Rating:** 6
**Confidence:** 4

**Summary:**

This manuscript argues that RL algorithms without considering the cost of hyperparameters tuning may lead negative effect to RL algorithms. The authors demonstrated their ideas via theoretical proving showing introducing hyperparamters in RL would definity induce extra cost. To address this issue, the authors propose new evaluation framework enabling fair comparisons. In addition, the authors performs additional experiments demonstrating the effectiveness of the proposed metrics.

**Strengths:**

The manuscript is well-written and mathematically solid. The structure is very clear and the claim is solid. It is appreciated that the author provides solid proves demonstrating the cost of black-box HPO in RL environment. It is appreciated that the authors not only show in ineffectiveness of the traditional HPO method in RL situation, but also delivers novel metrics for practical use.

**Weaknesses:**

The idea in this manuscript is quite interesting. However, there are some issues not clear.

The reviewer feels confusing about the Theorem 3.4, and the reviewer has another understanding. If Theorem 3.4 holds strongly, if the number of tuning parameters parameters $\Theta$ becomes large enough, will the value function converge to the optimal one? However, intuitively, when the number of tuning parameters increase, the model will be more difficult to converge, which shows the PAC guarantee is difficult to achieve under RL scenario, right?

In addition, in Theorem 3.5, ``with probability greater than 1/2'' does not make too much sense, right?

The number of parameters tuned range from 1 to 4 in the experiments, and each parameter only has 2-4 choices. Actually, the search space is not large enough (although not very small).  We suggest more hyperprameters for tuning proving in the experiment studies.

The experiment seems somehow confusing. The authors applies the proposed metrics for parameter tuning. However, it lacks comparison with other methods. Therefore, there are some difficulties claiming the proposed metrics are effective. It is suggested to consider some practical HPO methods as comparisons.

**Questions:**

See the weakness part above

**Alternative Position:**

Yes, and alternative positions are well-considered and named but not addressed

**Author Identification:**

No.

**Context:**

2

**Discussion:**

3

**Ethics:**

["NO or VERY MINOR ethics concerns only"]

**Position:**

Yes, the paper argues for or against a position related to machine learning.

**Support:**

3

**Thoroughness:**

3

---

### Note · Authors · 2025-08-18

**1-10 Additional Comments:**

We are genuinely disappointed by the quality of the reviews we received. They are even worse than the main conference. It's quite clear the reviews are rushed and most reviewers have not had the chance to read the paper very carefully.

**1-11 Submit Again:**

Unsure

**1-1 Submission Process:**

3

**1-2 Next Year:**

More exposure at the conference, e.g. best paper award, open debate, etc. The more drama the better.

**1-3 Future Development:**

Make the review process more rigorous and particularly make the criteria clear to the reviewers.

Invite senior researchers as reviewers, at similar levels to conference ACs.

(As an AC myself, I don't mind reviewing a few position papers in my area.)

**1-4 Interest:**

["Panel discussions with other position paper authors", "Structured debates on controversial topics", "Workshops for developing position papers"]

**1-4 Other Interest:**

A format where the authors tries to defend their position in front of a crowd, e.g. a poster session, and the audience tries to challenge their position.

**1-5 Thoughtful:**

3

**1-6 Supportive:**

8

**1-7 Technical Aspects Versus Position:**

8

**1-8 Gate Keeping:**

10

**1-9 Camera Ready Changes:**

1. Following the suggestion of reviewer oswF and kUYR, we will including the proofs to theorem 3.2, 3.4 and 3.5. and more preparation/explanation around them for audiences unfamiliar with statistical learning theory.
2. Following the suggestion of reviewer 4UpD, specifically mention the connection to meta-learning in section 5.1 --- this section is exactly about meta learning, where one first evaluate hyperparameters across a family of tasks and decide which ones to tune per task and which ones to fix across all tasks.

**3-1 Review Response1:**

oswF

**3-2 Reaction To Review1:**

Review oswF focuses almost entirely on the technical aspect of the paper, e.g. questioning the correctness of the theorems and requesting large scale experiments and comparison with other metrics.

Below we respond to some of the points brought up:

Q1: "if the number of tuning parameters parameters becomes large enough, will the value function converge to the optimal one?"
A1: Yes — Theorem 3.4 explicitly shows that the algorithm will converge to the optimal policy given sufficiently large
T. In practice, this also holds: one can always run the algorithm under each hyperparameter configuration and select the policy with the best performance.

Q2: "in Theorem 3.5, 'with probability greater than 1/2' does not make too much sense, right?"
A2: We respectfully disagree. The statement is a standard form of lower bound in PAC learning: it asserts that no black-box HPO algorithm can guarantee small regret with probability exceeding 1/2. Such “with probability at least 1/2” formulations are very common in impossibility results, and they correctly capture the inherent limitation of the problem.

Q3: "We suggest more hyperprameters for tuning in the experiment studies."
A3: This is a position paper. The experiments are intended as a demonstration of our proposed metrics rather than an exhaustive empirical study. The theoretical results already establish the general exponential scaling, which is independent of the number of hyperparameters in any particular experiment.

Q4: "However, it lacks comparison with other methods."
A4: Metrics, by design, are not directly comparable in the same way that algorithms are. The purpose of our proposed metrics is to provide a principled way to incorporate tuning cost into evaluation. Comparing “metric A” vs. “metric B” is not meaningful, because the goal is not to replace existing metrics, but to complement them with an HPO-aware perspective. Our experiments therefore illustrate how algorithm rankings can change under this new metric.

**3-3 Review Response2:**

4UpD

**3-4 Reaction To Review2:**

Below we respond to some of the points brought up:

Q1: "some may argue that reporting best-tuned results is still useful for comparing potential performance, especially if all methods are tuned equally"
A1: We agree that in certain settings—particularly when data collection is cheap or effectively unlimited—reporting best-tuned results can indeed be informative, since identifying the highest possible performance is the goal. However, our position is that in many practical domains where RL has not yet been successfully deployed (e.g., recommendation systems, healthcare, robotics), the cost of tuning is substantial and often prohibitive. In these scenarios, the lack of parameter-free or tuning-efficient algorithms is one of the major obstacles to adoption. Notably, if all methods are tuned equally, our proposed metrics still ensure fairness: the tuning cost is penalized uniformly, so relative comparisons remain valid.

Q2. "The authors also do not explore cases where tuning costs can be shared across tasks (e.g., via meta-learning or transfer HPO), which might reduce the impact of their main claim."
A2: We appreciate this observation. In fact, Section 5.1 of the paper directly addresses this idea. There, we simulate transfer by identifying insensitive hyperparameters whose values can be fixed across tasks based on averaged performance, while tuning only the sensitive ones on the new task. Our evaluation protocol naturally incorporates this setting: insensitive hyperparameters that transfer across tasks incur no additional penalty, and only the sensitive hyperparameters tuned online are penalized. This design allows our framework to reflect the benefits of transfer or meta-learning approaches, while still highlighting the cost of parameters that require fresh tuning.

**3-5 Review Response3:**

kUYR

**3-6 Reaction To Review3:**

Below we respond to some of the points brought up:

A1: "However, it is unclear whether their position also applies to offline RL."
Q1: Our theoretical results and position are framed in the online RL setting, where sample complexity and regret are the natural efficiency measures. That said, the challenges in offline RL are in some sense even more severe. Because unbiased policy evaluation is generally unavailable without online interaction, hyperparameter tuning becomes fundamentally ill-posed: one cannot reliably assess new configurations without resorting to additional assumptions or online rollouts. Thus, while our theorems do not directly apply, the underlying message—that ignoring tuning costs leads to misleading conclusions—remains valid, and arguably even stronger, in the offline RL setting.

A2: "Is developing parameter-free RL algorithms a feasible goal?"
Q2: We believe this is a feasible and timely goal. As discussed in lines 75–82, there is already emerging work on parameter-free RL in simplified settings, such as tabular environments and adversarial bandits. These pioneering efforts demonstrate proof-of-concept feasibility. Our position is that further investment in this direction is crucial, and our evaluation framework highlights why the community should prioritize parameter-free or tuning-efficient designs.

A3: "Could the authors provide a more detailed derivation of the proof in Section 3?"
Q3: The proofs are indeed straightforward for readers familiar with standard learning-theory techniques. However, we agree that providing complete derivations would improve accessibility. If space permits, we will include these full proofs in the camera-ready version, either in the appendix or supplementary material, to ensure clarity for all readers.

---

### Meta-Review · Area_Chair_C5Zt · 2025-09-16

**Rating:** 7
**Confidence:** 3

**Strengths:**

very timely topic and nicely formalized the connection between the hyperparameter tuning and sample complexity  in RL and distinguishing it from SL.

**Weaknesses:**

the writing could be improved

**Questions:**

why is this a position paper ?

**Thoroughness:**

3

---

### Decision · Program_Chairs · 2025-09-26

Reject